# Fragmentation in spin ice from magnetic charge injection

E. Lefrançois[1,2,4], V. Cathelin[2], E. Lhotel[2], J. Robert[2], P. Lejay[2], C.V. Colin[2], B. Canals[2], F. Damay[3], J. Ollivier[1], B. Fåk[1], L.C. Chapon[1,5], R. Ballou[2] & V. Simonet[2]

The complexity embedded in condensed matter fertilizes the discovery of new states of matter, enriched by ingredients like frustration. Illustrating examples in magnetic systems are Kitaev spin liquids, skyrmions phases, or spin ices. These unconventional ground states support exotic excitations, for example the magnetic charges in spin ices, also called monopoles. Here, we propose a mechanism to inject monopoles in a spin ice at equilibrium through a staggered magnetic field. We show theoretically, and demonstrate experimentally in the $Ho_2Ir_2O_7$ pyrochlore iridate, that it results in the stabilization of a monopole crystal, which exhibits magnetic fragmentation. In this new state of matter, the magnetic moment fragments into an ordered part and a persistently fluctuating one. Compared to conventional spin ices, the different nature of the excitations in this fragmented state opens the way to tunable field-induced and dynamical behaviors.

[1] Institut Laue Langevin, CS 20156, 38042 Grenoble, France. [2] Institut Néel, CNRS and Univ. Grenoble Alpes, 38042 Grenoble, France. [3] Laboratoire Léon Brillouin, CEA, CNRS, Univ. Paris-Saclay, F-91191 Gif-sur-Yvette, France. [4] Present address: Max Planck Institute for Solid State Research, Stuttgart D-70569, Germany. [5] Present address: Diamond Light Source Ltd., Harwell Science and Innovation Campus, Didcot OX11 0DE, UK. Correspondence and requests for materials should be addressed to V.S. (email: virginie.simonet@neel.cnrs.fr)

The spin ice state is the archetype of an exotic magnetic state that can be produced in the presence of geometric frustration and results from the inability of the system to find a unique ground state that minimizes its energy. It emerges in pyrochlore lattices of vertex-sharing tetrahedra when the magnetic moments are subjected to an effective ferromagnetic interaction and are constrained along the local ⟨111⟩ directions joining the corners to the center of each tetrahedron (diagonals of the cubic crystal)[1]. It is, classically, a macroscopically degenerate ground state, in which the spins locally obey the ice rule, with two spins pointing in and two spins pointing out of each tetrahedron (2I2O). Spin ice is the realization of a Coulomb phase[2]. Indeed, the ice rule is a local constraint that can be mapped on an emergent divergence free field, which is actually the local magnetization. This results in the famous pinch-point pattern in the magnetic diffuse scattering[3–5]. The spin ice elementary excitations are obtained by flipping one spin at the center of a pair of tetrahedra, resulting in 3 spins in–1 spin out (3I1O) and 1 spin in–3 spins out (1I3O) configurations. These excitations, called magnetic monopoles, can be described as two effective magnetic charges ($q = +1, -1$ for the 3I1O, 1I3O configurations respectively) that can be deconfined[6].

In the presence of dipolar interactions, and provided the charge density is large enough, the Coulomb-like interaction between monopoles can stabilize a monopole crystal (staggered 3I1O and 1I3O configurations) in an underlying spin-disordered manifold[7, 8]. The magnetic charge crystallization results in the fragmentation of the magnetization[8], a remarkable state where a single degree of freedom, the Ising magnetic moment, fragments thermodynamically into two parts, each part sustaining a different phase: an ordered antiferromagnetic phase, also interpreted as a crystal of magnetic charges (divergence full), and a disordered Coulomb phase (divergence free) with predominant ferromagnetic correlations.

A magnetic charge order, on top of degenerate spin configurations, was actually predicted earlier in spin models of dipolar ice on a kagome lattice (corner-sharing triangles)[9, 10] before being observed in a two-dimensional artificial spin ice, where the nanomagnets featuring the spins lie on a kagome lattice[11], and in a kagome oxide compound[12]. This peculiar state was finally shown to be associated with the fragmentation of the magnetization in an artificial kagome system[13]. The local ice rule in the kagome case actually corresponds to 2 in–1 out or 1 in–2 out configurations, so that alternating charges $q = \pm 1$ are present on each triangle. The charge crystal can therefore easily be stabilized in the kagome ice at variance with the pyrochlore spin ice where magnetic charges are a priori not present in the ground state. The fragmented state has nevertheless also been evidenced in a pyrochlore oxide compound[14] with more complex signatures due to quantum effects.

Here, we show that such fragmented state can be simply created in pyrochlore spin ices by injecting magnetic charges at equilibrium through a staggered magnetic field. By neutron scattering measurements, we provide evidence that this state is realized in the pyrochlore iridate compound $Ho_2Ir_2O_7$. In addition, signatures of unconventional excitations, equivalent to magnetic charges diffusing in a periodic potential, are observed, while plateau physics is identified in the magnetization as a function of an external magnetic field.

## Results

**Model.** A very appealing proposal is indeed to generate fragmentation of the magnetization by injecting magnetic charges in a controlled way, that is, with a tunable parameter such as a magnetic field, without starting from this required state of

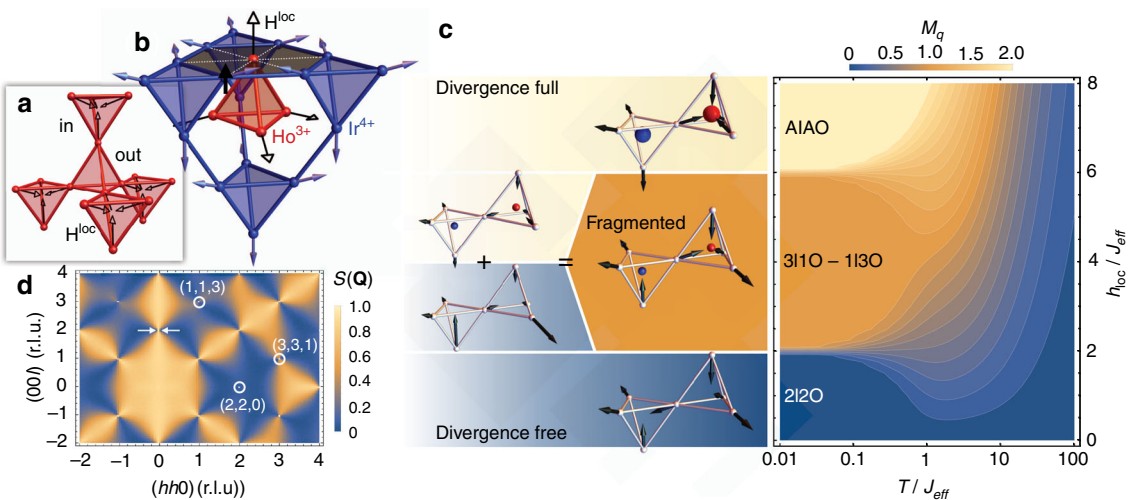

**Fig. 1** Staggered field in a pyrochlore lattice and magnetic fragmentation. **a** Pyrochlore lattice submitted to ⟨111⟩ staggered field represented by the *arrows* on each site, with in and out tetrahedra when this field points either inward or outward of the tetrahedra. **b** Structure of the pyrochlore iridate $Ho_2Ir_2O_7$ with the $Ho^{3+}$ ions (*red*) and the $Ir^{4+}$ ions (*blue*), both occupying a pyrochlore lattice. Each rare-earth is surrounded by a hexagon of six Ir nearest neighbors. When the iridium lattice orders magnetically in the AIAO phase, as shown on the *blue lattice*, a local magnetic field, perpendicular to the hexagon plane, hence aligned along the local ⟨111⟩ directions, is felt by the central rare-earth ion, as represented in **a**. **c** Phase diagram as a function of reduced temperature $T/\mathcal{J}_{eff}$ and local field $h_{loc}/\mathcal{J}_{eff}$, defined through the charge order parameter $M_q = \left\langle \left| \frac{1}{N} \sum_{\alpha=1}^{N} \Delta_\alpha q_\alpha \right| \right\rangle$ where $N$ is the number of tetrahedra and $\Delta_\alpha = +1$ (−1) for in (out) tetrahedra carrying a charge $q_\alpha$. As $h_{loc}/\mathcal{J}_{eff}$ increases, the ground state changes from the spin ice ($M_q = 0$, *blue*), to the AIAO state ($M_q = 2$, *yellow*), going through the charge crystal intermediate phase which fragments ($M_q = 1$, *orange*). The fragmentation can be described by the splitting of the pseudo-spin $\sigma = \pm 1$: the divergence free fragment on a tetrahedron can then be written as a state with four spins such that $\sigma_{\langle |q| \rangle = 0} = (\pm 1/2, \pm 1/2, \pm 1/2, \text{ and } \mp 3/2)$, while the divergence full fragment that carries the magnetic charge corresponds to an AIAO configuration with half of the moment so that $\sigma_{\langle |q| \rangle = 1} = (\pm 1/2, \pm 1/2, \pm 1/2, \pm 1/2)$[8]. All the spin and charge configurations are shown in the left panel of **c** at $T = 0$. **d** Magnetic scattering function $S(\mathbf{Q})$, with $\mathbf{Q}$ the scattering vector, calculated in the fragmented phase. It shows a diffuse pattern with pinch points (evidenced by *arrows*) together with the magnetic Bragg peaks (*highlighted by circles*) characteristic of the AIAO ordered state, but whose intensity (proportional to the magnetic moment squared) is a quarter of that expected when the whole magnetic moment is ordered

monopole crystal, which imposes strong constraints on the Hamiltonian[8]. However, in a pyrochlore spin ice, any external magnetic field does not act homogeneously on the magnetic moments of the unit cell, since the magnetic moments inside a tetrahedron point in different $\langle 111 \rangle$ directions. When the field is applied along one of the $\langle 111 \rangle$ directions, along which the pyrochlore lattice can be viewed as a stacking of triangular and kagome planes, only a partial fragmentation occurs (within the kagome planes). This leads to the so called kagome ice phase[15, 16] in which a charge crystal is stabilized in the kagome planes through a 2 in–1 out, 1 in–2 out rule, similar to artificial spin ices. However, no magnetic charges are actually present in the tetrahedral units.

We propose below an approach to produce the fragmentation process in the whole pyrochlore lattice, even in the absence of dipolar interactions, by considering a magnetic field exerted along the local directions of the magnetic moments. As depicted in Fig. 1a, if the sign of this field alternates, pointing inward for a given tetrahedron (called in) and outward for the neighboring tetrahedra (called out), we show that this staggered field competes with the spin ice state. This results, over a large field range, in a fragmented ground state supporting unconventional excitations.

Consider the nearest-neighbor spin ice Hamiltonian in the presence of a local magnetic field:

$$\mathcal{H} = -\mathcal{J} \sum_{\langle i,j \rangle} \mathbf{S}_i \cdot \mathbf{S}_j - g\mu_0\mu_B \sum_i \mathbf{H}_i^{loc} \cdot \mathbf{S}_i$$
$$= \mathcal{J}_{eff} \sum_{\langle i,j \rangle} \sigma_i \sigma_j - h_{loc} \sum_i \sigma_i \tag{1}$$

Where, $\mathbf{S}_i$ is the $i$th magnetic moment pointing along its local trigonal direction and interacting with its nearest neighbors via a ferromagnetic interaction $\mathcal{J}$. $\sigma_i = \pm 1$ is the corresponding Ising pseudo-spin which is equal to +1 (−1) when the moment points inward (outward) the tetrahedron. $\mathcal{J}_{eff} = \mathcal{J}/3S^2 > 0$ is the effective nearest-neighbor interaction[17]. $\mathbf{H}_i^{loc}$ is the staggered magnetic field described above, aligned along the $\langle 111 \rangle$ direction of the $i$th site, and $h_{loc} = g\mu_0\mu_B H^{loc}S$.

Minimization of Eq. (1) on a single tetrahedron, together with Monte Carlo simulations on a pyrochlore lattice (Fig. 1c and Supplementary Fig. 7), gives a succession of ground states with a different charge order parameter, depending on $h_{loc}/\mathcal{J}_{eff}$. At low fields, the ground state is the 2I2O spin ice state ($\langle |q| \rangle = 0$). In the opposite limit of large fields, the stabilized state is the all in–all out (AIAO) antiferromagnetic order (charge crystal with $\langle |q| \rangle = 2$), in which all the spins of a given tetrahedron point either inward or outward, following the staggered field.

More exotic physics appear in the intermediate regime $2 < h_{loc}/\mathcal{J}_{eff} < 6$ (*orange* region in the phase diagram of Fig. 1c): the competition between $\mathcal{J}_{eff}$ and $h_{loc}$ selects 3I1O and 1I3O configurations on the in and out tetrahedra respectively. It thus stabilizes a charge crystal, with $\langle |q| \rangle = 1$, and leads to the fragmentation process[8], characterized by a staggered magnetization at half of the magnetic moment. Another important signature is the coexistence of diffuse scattering with a pinch-point pattern and AIAO Bragg peaks in the computed scattering function (Fig. 1d). Here, the fragmented state does not result from the presence of the long-range monopole Coulomb energy but from the injection at equilibrium of magnetic charges into a spin ice state by a staggered magnetic field.

**Experimental evidence for fragmentation**. A prominent system to realize this model is the pyrochlore iridate family, of formula $R_2Ir_2O_7$ in which R is a rare-earth element. In these compounds, the iridium and rare-earth ions lie on two interpenetrated pyrochlore lattices (Fig. 1b). It was shown in pyrochlore iridates where R = Nd, Eu, and Tb, that the iridium sublattice orders in an AIAO magnetic arrangement[18–20], simultaneously with a metal insulator transition[21] ($T_{MI} = 30$–140 K). This arrangement generates a molecular field oriented along the local $\langle 111 \rangle$ directions on the rare-earth sublattice[20], and thus plays exactly the same role as our staggered magnetic field $h_{loc}$ in Eq. (1). The fragmentation process described above can thus be realized in the pyrochlore iridates, provided the rare-earth ions have a local easy axis anisotropy along the $\langle 111 \rangle$ directions and interact through a ferromagnetic interaction. This is the case of Ho and Dy ions in pyrochlore compounds with nonmagnetic ions instead of $Ir^{4+}$, and where spin ice physics is observed at Kelvin temperatures[1, 22–25]. In the present article, we focus on the Ho pyrochlore iridate.

Our measurements were performed on a high-quality polycrystalline sample of $Ho_2Ir_2O_7$. As previously reported[21], the magnetization exhibits a zero field cooled-field cooled (ZFC-FC) bifurcation at $T_{MI} = 140$ K pointing out the magnetic ordering of the Ir sublattice (Supplementary Fig. 1). Below this temperature, the iridium sublattice starts to produce a molecular field on the holmium sites, which manifests as the rise of magnetic Bragg peaks below about 80 K in the neutron diffraction pattern. Due to the very weak moment of the iridium, these Bragg peaks are mainly characteristic of an AIAO magnetic ordering of the holmium sublattice (Fig. 2a). This is the ordering expected in our model when the temperature and the staggered magnetic field are large with respect to the effective interaction. From the temperature evolution of the refined $Ho^{3+}$ ordered magnetic moment, strongly increasing when decreasing the temperature, we could extract the value of the molecular field $H^{loc}$ estimated to $0.725 \pm 0.05$ T below 40 K (Fig. 2b). This was achieved following the procedure of ref. [20] and using the crystal electric field parameters of the $Ho^{3+}$ ion deduced from inelastic neutron scattering (INS) experiments (Supplementary Notes 1 and 2 and Supplementary Figs. 2–4).

When decreasing further the temperature, the Ho-Ho magnetic interactions start to become relevant. The ordered moment finally saturates below 2 K, ending with a value of $5 \pm 0.06 \, \mu_B$ at 50 mK. This value is only half the magnetic moment of the $Ho^{3+}$ ions in their ground state doublet (Supplementary Note 2). A diffuse scattering signal is actually present and accounts for the missing moment, as seen in Fig. 2c. It has broad maxima at the scattering vectors $Q \sim 0.7$, 1.6, and 2.7 $Å^{-1}$, evidencing additional spin correlations characteristic of the spin ice diffuse scattering[23, 25], resulting from the powder average of the pinch-point pattern observed on single crystals[5]. This coexistence of a spin ice diffuse scattering with an AIAO ordering, whose ordered magnetic moment is half of the total moment, is remarkably well reproduced in the calculations (Fig. 2d) and is the signature that a fragmentation process occurs in $Ho_2Ir_2O_7$ below 2 K. Finally, the calculated temperature dependence of the ordered moment using Eq. (1) gives $\mathcal{J}_{eff} = 1.4 \pm 0.1$ K and $h_{loc}/\mathcal{J}_{eff} = 4.5 \pm 0.25$ (Fig. 2b), which places $Ho_2Ir_2O_7$ deep into the fragmentation regime according to Fig. 1c. It corresponds to a local field $\mu_0 H^{loc} = 0.94 \pm 0.02$ T, in the same range as the molecular field estimated from the high temperature behavior.

We have thus demonstrated that the thermodynamical fragmentation of the magnetic moment under the influence of a local staggered field predicted in our model appropriately describes the unexpected properties of $Ho_2Ir_2O_7$. In the following, we probe the macroscopic magnetic properties of $Ho_2Ir_2O_7$ to explore the excitations emerging from this fragmented state.

**Dynamics in the fragmented state**. In the fragmented regime, typically below 2 K, the alternative (AC) susceptibility displays a frequency dependence, which can be described by a thermally

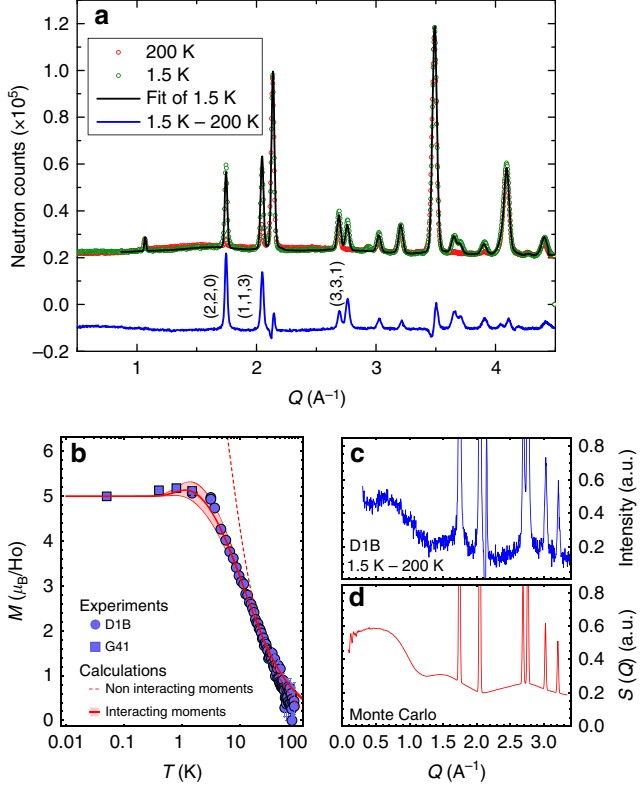

**Fig. 2** Magnetic ordering and diffuse scattering in $Ho_2Ir_2O_7$. **a** Neutron diffractograms recorded at 200 K (*red dots*) and 1.5 K (*green dots*) showing the rise of magnetic Bragg peaks. The 1.5 K Rietveld refinement (*black line*) using a **k** = **0** propagation vector yields an AIAO magnetic order for the holmium sublattice. The difference between the 1.5 and 200 K data (*blue line,* shifted for clarity) enhances the magnetic Bragg peaks. **b** Temperature evolution of the refined $Ho^{3+}$ magnetic moment, in a semi-logarithmic scale, measured with two diffractometers (D1B and G4.1). The *red dashed curve* is the calculated ordered magnetic moment induced by the temperature dependent molecular field created by the Ir magnetic ordering. Below 20 K, the calculated and experimental curves depart from each other due to the presence of Ho-Ho magnetic interactions. The *red full curve* is obtained from Monte Carlo calculations (Eq. (1)) with $\mathcal{J}_{eff} = 1.4$ K and $h_{loc}/\mathcal{J}_{eff} = 4.5$, the *colored area* representing the calculated values compatible with the experimental *error bars* (within symbol size). These calculations allow to account for the saturation of the ordered moment below 2 K due to fragmentation. The departure above 40 K between the calculated and experimental curves is attributed to the decrease of the Ir molecular field. **c** Evidence for diffuse scattering at 1.5 K, from the difference between the 1.5 and 200 K diffractograms (the negative intensity is due to an imperfect subtraction of a strong nuclear peak caused by the lattice parameter variation). **d** Powder average magnetic scattering function $S(Q)$ from Monte Carlo calculations at $T/\mathcal{J}_{eff} = 0.05$ with $h_{loc}/\mathcal{J}_{eff} = 4.5$

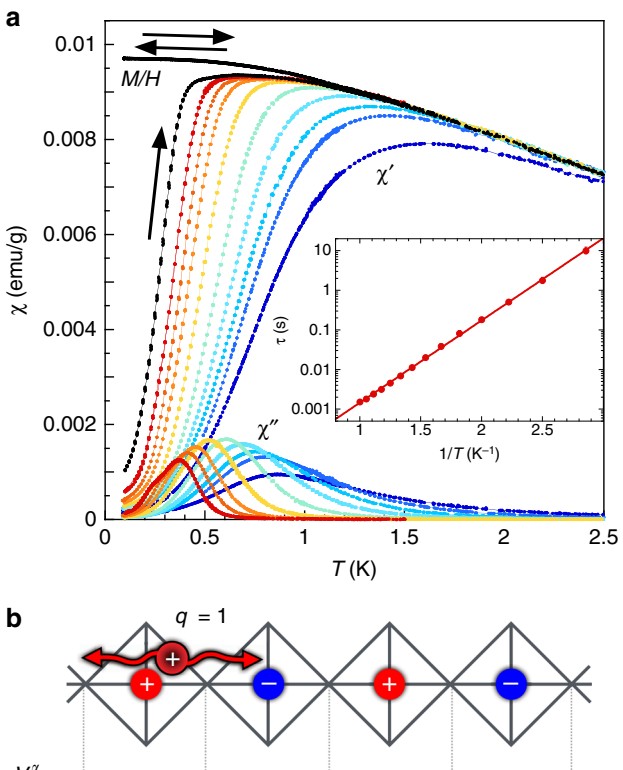

**Fig. 3** Dynamical properties and diffusion of the excitations. **a** Low temperature dependence of the ZFC-FC magnetization performed in 5 mT below 4 K (*black symbols*). The two curves depart below about 1.5 K. On the same graph are shown the real part $\chi'$ and imaginary part $\chi''$ of the AC susceptibility vs $T$ at several frequencies $f$ from 0.011 Hz (*red*) to 570 Hz (*blue*) with the intermediate frequencies 0.057, 0.21, 1.11, 5.7, 21, 57 and 111 Hz. Inset: $\tau$ vs $1/T$, where $\tau = 1/2\pi f_{peak}$ and $f_{peak}$ is the maximum of $\chi''(f)$ measured at constant temperature $T$. The line is a fit to an Arrhenius law $\tau = \tau_0 \exp(E/T)$ where $\tau_0 = (1.2 \pm 0.2) \times 10^{-5}$ s and $E = 4.8 \pm 0.1$ K. **b** Two-dimensional representation of a string of tetrahedra on which an elementary excitation carrying a charge $q = +1$ (*red disc with black border*) diffuses in the fragmented regime Spatial periodic potential $V_q^\alpha$ felt by the diffusing charge and induced by the underlying static charge crystal, where $\alpha$ refers to the tetrahedron type: $\alpha =$ in carries an ordered charge $q_o^{in} = +1$ (*red*) and $\alpha =$ out carries $q_o^{out} = -1$ (*blue*)

through a local interaction between the diffusive charge and the ordered charges. The energy of a single excitation is then given by

$$E_q^\alpha = 2\mathcal{J}_{eff} + V_q^\alpha, \qquad (2)$$

with $V_q^\alpha = (4\mathcal{J}_{eff} - h_{loc})q_o^\alpha q$, and where $q_o^\alpha = -1$, +1 is the ordered charge on a tetrahedron with $\alpha =$ out, in. For $h_{loc}/\mathcal{J}_{eff} = 4.5$, corresponding to the $Ho_2Ir_2O_7$ case, the periodic potential is alternatively repulsive (attractive) for a diffusing charge interacting with an ordered charge of opposite (same) sign (Fig. 3b). The relaxation timescale is then given by the highest energy barrier $E = h_{loc} - 2\mathcal{J}_{eff} \simeq 3.5$ K, rather close to the experimental value. Long-range dipolar interactions[17], as well as more complex diffusion mechanisms (Supplementary Note 4 and Supplementary Figs. 8 and 9), could significantly affect the

activated process above an energy barrier $E \approx 4.8$ K (Fig. 3a, Supplementary Note 3 and Supplementary Figs. 5 and 6). It is associated with a freezing in the ZFC-FC magnetization below 1 K, where a strong irreversibility is observed. Similar slow dynamics are observed in spin ice materials[26, 27], where they are induced by the diffusion of monopoles, with a relaxation time-scale inversely proportional to the monopole density[17, 28, 29]. In the present case, the dynamics can also be understood by the diffusion of the fractional excitations, of energy $2\mathcal{J}_{eff}$ and charge $q = \pm 1$, emerging from the fragmented Coulomb phase. But a marked difference with the spin ice is that these excitations are diffusing in a periodic potential $V_q^\alpha$ that depends on the local field. It is created by the underlying charge crystal fragment

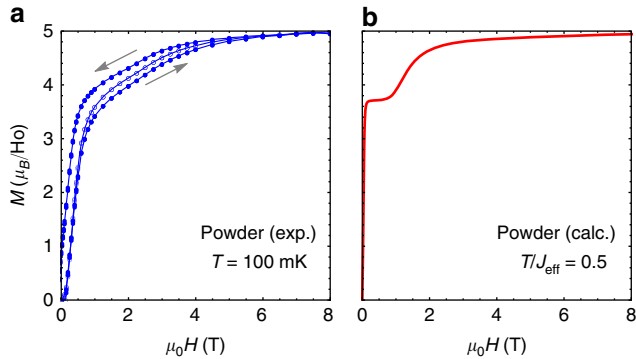

**Fig. 4** Magnetization, measurements and calculations. **a** $M$ vs $H$ measured at 110 mK. The *empty symbols* show the first magnetization curve and the *filled symbols* the magnetization curves obtained afterwards by ramping the field up and down. The small hysteresis opening is due to the existence of slow dynamics at low temperature. **b** Calculated $M$ vs $H$ for $h_{loc}/\mathcal{J}_{eff} = 4.5$ and $T/\mathcal{J}_{eff} = 0.5$, performing a powder averaging. The anomalies, mimicking the experimental curve, feature the existence of an exotic field-induced behavior exhibiting magnetization plateaus

relaxation timescale and improve the agreement with experiments.

**Field-induced behavior.** The application of an external uniform magnetic field is expected to promote exotic phases. In $Ho_2Ir_2O_7$, isothermal magnetization curves at very low temperature do exhibit an unconventional behavior (Fig. 4a): the magnetization first rises abruptly to reach a plateau-like feature at $\approx 3.5 \, \mu_B/Ho$, before increasing again above 2 T up to $5 \, \mu_B$ at 8 T (which is the expected value for Ising Ho moments along the $\langle 111 \rangle$ directions[30]). This fine structure of the magnetic isotherm is qualitatively reproduced by Monte Carlo calculations in the fragmented state using a powder average (Fig. 4b). It actually reflects the rich plateau physics emerging from the fragmented ground state. Several field-induced phases are stabilized depending on the orientation of the external magnetic field, for example successive fragmented kagome ice states (Supplementary Note 5 and Supplementary Fig. 10). Further experimental and theoretical works are required to probe these exotic states in more details[15, 16, 31, 32].

**Discussion**

The introduction of a staggered magnetic field in the nearest-neighbor spin ice Hamiltonian on the pyrochlore lattice thus allows, through magnetic charge injection, to stabilize a fragmented state, and offers a rich playground to study the various facets of spin ice and monopoles physics in a renewed perspective. A step further would be to account for dipolar interactions in the Hamiltonian, which might affect the actual ground state and the dynamics at very low temperature.

We have shown that, in the pyrochlore iridate $Ho_2Ir_2O_7$, the ratio between the rare-earth interactions and the staggered field induced by the iridium moment places the system deeply in the fragmented phase. Experimentally, it would then be very appealing to explore the full fragmentation phase diagram and the associated exotic dynamics by varying the ratio between rare-earth interactions and the staggered local field. This can be done by changing the rare-earth element. Preliminary measurements on the Dy pyrochlore iridate suggest that fragmentation also occurs in this compound, and hopefully, this system will be located at a different $h_{loc}/\mathcal{J}_{eff}$. This phase diagram mapping should also be achieved more systematically by applying an

external isostatic pressure. Finally, the model we have developed is perfectly equivalent to the case of antiferromagnetically-coupled collinear Ising spins on a pyrochlore lattice submitted to a uniform external field, which can be easily varied. It thus opens the route to the tuning of fragmentation in new systems.

**Methods**

**Calculations.** Thermodynamical quantities as well as structure factors have been calculated through Monte Carlo simulations of the spin model described by Eq. (1). A single spin-flip Metropolis update has been combined to a loop algorithm preventing the system from being trapped into a small region of the configuration space at low temperature. The simulations were performed on $16 \times L^3$ lattice sites with $L = 8, 12$ (Figs. 1d and 4b) and 24 (Figs. 1c and 2d) with periodic boundary conditions. In total, $10^4$ hybrid Monte Carlo steps were used for thermalization, while the measurements were computed over $N$ steps, where $N$ is adapted to ensure stochastic decorrelation between measurements (typically, from $N = 10^4$ at high temperature to $10^7$ at low temperature). The powder magnetization of Fig. 4b, averaged over 800 random field directions, has been computed at sufficiently high temperature to preserve the efficiency of the used algorithm.

**Synthesis.** Polycrystalline samples of $R_2Ir_2O_7$, with R = $Ho^{3+}$ ($4f^{10}$, $S = 2$, $L = 6$, $J = 8$, $g_J = 5/4$, $g_J J = 10 \, \mu_B$) of high quality were synthesized by a mineralization process. The starting reagents $R_2O_3$ and $IrO_2$ were mixed together with a small amount of KF flux and pressed into a pellet. After being placed in a Pt crucible, they were submitted to a heat treatment under air atmosphere: 200 °C .h$^{-1}$ until a plateau of 120 h at 1110 °C and a decrease to room temperature at 200 °C .h$^{-1}$. The pyrochlore iridates crystallize in the $Fd\bar{3}m$ cubic space group, with the O occupying the 48f and 8b Wyckoff positions, the rare-earth and the Ir occupying the 16d and 16c positions respectively. The structure and quality of the samples were checked by X-ray diffraction, with no trace of parasitic phases. The lattice parameter and the $x$ coordinate of the 48f O were found at room temperature equal to 10.1790(2) Å and 0.343(2) Å.

**Magnetization measurements.** The temperature and field dependences of the magnetization of the $Ho_2Ir_2O_7$ powder sample down to 2 K were measured using Quantum Design VSM and MPMS® SQUID magnetometers and are shown in Supplementary Fig. 1. The magnetization vs temperature exhibits a ZFC-FC bifurcation at $T_{MI} = 140$ K pointing out the ordering of the Ir sublattice concomitant with the metal-insulator transition. The magnetic isotherms, linear at high temperature, are approaching a plateau of $\approx 5 \, \mu_B/Ho$ at 2 K, which reflects the strong Ising anisotropy of $Ho^{3+}$.

Very low temperature magnetization measurements were performed on SQUID magnetometers equipped with a dilution refrigerator developed at the Institut Néel[33]. The powder samples were crushed and mixed with Apiezon grease in a copper pouch to ensure a good thermal contact between the copper sample holder and the sample. AC susceptibility measurements were performed with an AC field of 0.1 or 0.23 mT. Demagnetization corrections were performed assuming an average demagnetization factor of 0.1 (cgs units) estimated from the shape of the copper pouch.

**Neutron diffraction.** The CRG-D1B diffractometer at the Institut Laue Langevin (ILL) (wavelength $\lambda = 2.52$ Å) was used in the 1.5–300 K temperature range and the G4.1 diffractometer at the LLB ($\lambda = 2.427$ Å) was used for the lower temperature range. It is equipped with a Cryoconcept-France HD dilution refrigerator (100 μW at 100 mK), and fitted with vanadium shields. The sample was put in a vanadium cell in 14 bars of He gas. Rietveld refinements were carried out with the FULLPROF program[34].

**Inelastic neutron scattering.** The magnetic excitations in $Ho_2Ir_2O_7$ were studied by INS on the same polycrystalline sample as the one used for neutron diffraction and magnetometry measurements. The experiments were carried out at the ILL on the IN6 and IN4 neutron spectrometers. On the IN6 cold neutron time-of-flight spectrometer, the incoming neutron wavelength (energy) was 5.1 Å (3.145 meV). The energy resolution for elastic scattering was $\delta E \approx 70$ μeV. On the IN4 thermal neutron time-of-flight spectrometer, the incoming neutron wavelengths (energies) used were 2.44 Å (13.74 meV), 1.3 Å (48.4 meV), and 0.9 Å (101 meV). The energy resolution for elastic scattering in the respective settings was $\delta E \approx 1$, 3.75 and 9.5 meV. Results are shown and discussed in Supplementary Notes 1–2.

**Data availability.** All relevant data are available from the authors. Neutron scattering data performed at the ILL are available at the doi: 10.5291/ILL-DATA.5-31-2406 for neutron diffraction data and doi: 10.5291/ILL-DATA.4-01-1478 for INS data.

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

## Acknowledgements

We acknowledge A. Hadj-Azzem and J. Balay for their help in the compound synthesis. We thank C. Paulsen for allowing us to use his SQUID dilution magnetometers and P.C. W. Holdsworth, S. Petit and S. de Brion for fruitful discussions. VC and ELh acknowledge financial support from ANR, France, Grant No. ANR-15-CE30-0004. R.B., B.C., C.C., F.D., P.L., E.L.h., J.R. and V.S. acknowledge financial support from ANR, France, Grant No. ANR-13-BS04-0013-01.

## Author contributions

The sample synthesis was performed by P.L. R.B., V.S., L.C.C., E.Le. and E.Lh. conceived and designed the experiments. Neutron scattering experiments were carried out by E.Le., V.S., R.B., E.Lh., L.C.C., F.D., C.V.C., J.O. and B.F. The magnetometry measurements were carried out by E.Le. (standard), V.C. and E.Lh. (very low temperature). The calculations were performed by J.R. with the support of B.C. The data analysis was done by E.Le., V.C., E.Lh., J.R., R.B., V.S. and F.D. The article was written by E.Lh., V.S., J.R. and R.B. with feedback from all authors.

## Additional information

**Competing interests:** The authors declare no competing financial interests.

