## [Peer Review File · Nature Communications]

Reviewers' Comments:

Reviewer #1 (Remarks to the Author):

This paper presents a combined experimental and theoretical study on a new exotic phase in pyrochlore spin ice $\text{Ho}_2\text{Ir}_2\text{O}_7$. In this special compound, long-range magnetic ordering of the Ir-moment (so-called all-in-all-out) creates a local molecular field on the Ho-spins which is equivalent to a staggered magnetic field. As the author pointed out, an intriguing result of the staggered magnetic field on spin-ice is to stabilize a monopole-crystal phase in which disordered spins coexist with long-range ordered magnetic charges. Monte Carlo simulations based on short-range spin-spin interaction and an external staggered magnetic field well reproduce many of the experimental results.

This is a very interesting work, reporting the experimental detection of a new state of matter. By combining the Ir-atom (well studied all-in-all-out order) with Ho-atom (canonical spin-ice) in the pyrochlore structure, this work also provides a ingenious example of materials-by-design approach.

Here I have a few minor comments about the paper.

(1) I understand the authors want to focus the references only on pyrochlore related spin ice systems. However, the existence of such novel phase (Neel ordering of magnetic charge coexisting with disordered spins) was first pointed out in the context of kagome spin ice. So instead of citing Refs. 13, 14 (which are about the kagome-ice phase in pyrochlore), I think the authors should cite the following two papers: PRB 80, 140409(R) (2009) and PRL 106, 207202 (2011), where the novel partially ordered phase was first proposed in the context of spin ice.

It might be also worth pointing out that such monopole-crystal phase (with disordered spins) was also observed in artificial kagome spin ice and a recent three-dimensional layered spin ice compound; see, e.g. Nature 500, p.553 (2013) and Nature Communication 7, 13842 (2016).

I don't think citing these relevant works would diminish the novelty or originality of this nice work.

(2) At the lowest temperature, the long-range dipolar interaction would most likely lift the quasi-degeneracy of the monopole-crystal phase and induces long-range spin order. Can the

authors comment on what kinds of long-range magnetic order is expected ?

(3) This is purely optional. Another signature of the monopole-crystal phase is the coexistence of pinch-point and Bragg peaks in structure factor, as the authors pointed out in Fig. 1(d). What is the main difficulty experimentally to detect this ? (single-crystal not possible ?)

In summary, I enthusiastically recommend the publication of this work in Nature Communications. And I think the paper would be more complete if the authors can address the above comments.

Reviewer #2 (Remarks to the Author):

This paper explores the influence of a staggered magnetic field on spin ice, and shows that it leads to a “fragmented” phase in which there is partial long range order of the all-in-all-out type, as well as short range correlations from the part of the moments involved in spin ice configurations (2-in-2-out).

The authors show that this description can reproduce the powder neutron diffraction data on $\text{Ho}_2\text{Ir}_2\text{O}_7$, in which the Ir sublattice orders at high temperature into an all-in-all-out state, and creates a staggered molecular field on the ferromagnetically-coupled Ho ising sublattice. These are the ingredients necessary to induce fragmentation, as described fully in the SI.

The results are interesting from the perspective of fragmentation in spin ice, a relatively new idea which has seen some success in describing certain pyrochlore materials as well as artificial spin ice. This is a new take on those ideas, and broadens the fragmentation topic to include pyrochlore materials with two distinct magnetic sublattices. The work is also novel from the perspective of understanding the magnetic details of $\text{Ho}_2\text{Ir}_2\text{O}_7$, which is part of a hotly studied class of pyrochlore materials. The coexistence of magnetic Bragg peaks indicating long range order and spin-ice-like diffuse scattering in this material is unusual, and the fact that these details can be understood quantitatively within this model is remarkable. Thus, I think there is enough novelty here to warrant publication in Nature Communications. My only criticism on this point is that it seems that if the goal is to have a tunable “charge injection” mechanism in spin ice, it would be more tractable to start with the AFM ordered pyrochlore (AIAO ordered) and apply a uniform magnetic field (the authors point out at the very end of the manuscript that this is an equivalent model). Nonetheless, the authors do suggest tuning the interactions in this staggered field model using external pressure, which I think could work. Their other suggestion, “tuning” the rare earth ion, is a misnomer, as there seems to only be a couple of choices in the rare earth iridates that give the necessary FM coupling between Ising moments on the rare earth sublattice (Ho and Dy) while also having an insulating ground state and ordered Ir moments. This is a subjective point of view, however, and I don’t feel that any changes to the manuscript necessarily need to be

made regarding this.

Overall I believe that the results are robust and the analysis is sound. There is a lot of information in the SI, but I believe it is correctly placed (i.e., it does not need to be in the main part of the manuscript). However, I do have some suggestions for improvement of the main manuscript, listed in point form below.

- From a clarity standpoint, I think it would be better to use capital letters for the abbreviations like “all-in-all-out” (aiao). Otherwise, the word “aiao” which appears several times could be misunderstood as a typo or some unusual word unknown to the readers. AIAO would make it clear it is an acronym.

- In figure 1, on the left side of panel c), it is not clear whether the x-axis relates to temperature (as it does on the right side). The colors fading out seem to suggest this, but it is not labelled on that axis. Some text relating to this point would be helpful.

- Figure 2 b) shows the ordered magnetic moment as a function of temperature in $\text{Ho}_2\text{Ti}_2\text{O}_7$, extracted from neutron powder diffraction. The authors argue there is a slight overshoot in the data around 1K, which is consistent with the theoretical model. This is not actually clear in the data, especially since there are no errorbars on those points. It could easily be a plateau. The errorbars should be added, and the wording that describes the possible overshoot should be softened.

- In Figure 3, the specific frequencies of the ac susceptibility data need to be indicated in some way (as a legend, labels directly on the plot, or in the caption with specific start and end points as well as the step size in f).

- small points: Is the instrument called G41 or G4.1? It is labelled as G41 in the inset of Fig. 2, but in at least two places in the text it is called G4.1. Secondly, in the methods section the word “gaz” should be “gas”.

In summary, I think this is very nice work with a new take on fragmentation in spin ice, that also helps to explain experimental features of a pyrochlore material of current interest, $\text{Ho}_2\text{Ir}_2\text{O}_7$. Once the points above are addressed, I think it will be suitable for publication in nature communications.

Reply to Reviewers

Reviewer 1

We thank Reviewer 1 for the interest he/she shows for our study.

(1) I understand the authors want to focus the references only on pyrochlore related spin ice systems. However, the existence of such novel phase (Neel ordering of magnetic charge coexisting with disordered spins) was first pointed out in the context of kagome spin ice. So instead of citing Refs. 13, 14 (which are about the kagome-ice phase in pyrochlore), I think the authors should cite the following two papers: PRB 80, 140409(R) (2009) and PRL 106, 207202 (2011), where the novel partially ordered phase was first proposed in the context of spin ice.

It might be also worth pointing out that such monopole-crystal phase (with disordered spins) was also observed in artificial kagome spin ice and a recent three-dimensional layered spin ice compound; see, e.g. Nature 500, p.553 (2013) and Nature Communication 7, 13842 (2016).

I dont think citing these relevant works would diminish the novelty or originality of this nice work.

Answer: Indeed, we had focused the introduction on the case of the pyrochlore systems, just mentioning kagome systems in the specific case where fragmentation had been identified. We agree that charge ordering was first pointed out in the context of kagome and artificial spin ice systems, and we have changed the introduction to mention it.

We have kept the references to the observation of kagome ice phases in the pyrochlore systems (which were discussed before artificial spin ice systems, without realizing that they could be described within a charge ordering formalism). Actually, this is important in the context of our study that such phases can be stabilized in pyrochlore systems when an external magnetic field is applied: We also predict kagome ice phases in our model with a staggered field and find some signature of their realization through a plateau-like behavior in the magnetization curve.

(2) At the lowest temperature, the long-range dipolar interaction would most likely lifts the quasi-degeneracy of the monopole-crystal phase and induces long-range spin order. Can the authors comment on what kinds of long-range magnetic order is expected ?

Answer: We agree that long-range dipolar interactions would most likely lift the degeneracy of the monopole-crystal phase. This is the case in kagome ice. This is also the case in pyrochlores in the “conventional” dipolar spin ice (i.e. in absence of charge crystallization and fragmentation) where at low temperature the system is theoretically expected to order.

In the theoretical model of fragmentation proposed by Brooks-Bartlett et al., no ordering of the spin configurations was mentioned down to zero temperature, but this might be due to the fact that the dumbbell model does not fully account for the dipolar interactions. However, in the article of Borzi et al. (Phys. Rev. Lett. 111, 147204 (2013) - Ref 7 of the present manuscript), the

authors study theoretically the dipolar spin ice model, at different monopole densities, and they indeed find that, at low temperature, below the charge ordering temperature, a transition towards an ordered spin state occurs.

In our model with a staggered field, a keypoint is that we do not need to account for long range interactions to stabilize the charge ordering and the fragmentation. Consequently, in the model, the spin system is not ordered even at zero temperature, as in our experiments performed down to 50 mK. Accounting for long-range dipolar interactions may however also change the ground state of the system at sufficiently low temperature. We can speculate that some specific 3 in – 1 out / 1 in – 3 out spin configurations are more favorable with respect to long-range interactions, so that ordering may also be achieved in this case, but we cannot say at the moment which kind of long-range magnetic order would be stabilized.

We have not been into the details of this discussion in the manuscript, because it goes far beyond the results we present here. But we have added a sentence in the discussion suggesting that a different ground state may exist at very low temperature when considering the dipolar interactions.

(3) This is purely optional. Another signature of the monopole-crystal phase is the coexistence of pinch-point and Bragg peaks in structure factor, as the authors pointed out in Fig. 1(d). What is the main difficulty experimentally to detect this ? (single-crystal not possible ?)

The monopole crystal and the fragmentation are indeed characterized by the coexistence of pinch points and Bragg peaks in the magnetic scattering function. Unfortunately, in Ho and Dy pyrochlore iridates, no single crystals are available so that the pinch point structure cannot be resolved. However, a rich neutron scattering literature exists in spin ice, especially for Ho based compounds (see for example Ref. 23 and 25), which allows to identify the signature of spin ice correlations in the diffuse scattering of powder samples. The diffuse scattering we observe perfectly matches these previous measurements, so that we can confidently conclude that it is associated with correlations which would have lead to a pinch point pattern for a single crystal.

We have added a sentence when discussing the diffuse scattering to precise that this diffuse scattering correspond to the pinch point pattern averaged for a polycrystalline sample.

Reviewer 2

We thank Reviewer 2 for his/her careful reading and for the interest he/she shows in our results.

(1) My only criticism on this point is that it seems that if the goal is to have a tunable “charge injection” mechanism in spin ice, it would be more tractable to start with the AFM ordered pyrochlore (AIAO ordered) and apply a uniform magnetic field (the authors point out at the very end of the manuscript that this is an equivalent model). Nonetheless, the authors do suggest tuning the interactions in this staggered field model using external pressure, which I think could work. Their other suggestion, “tuning” the rare earth ion, is a misnomer, as there seems to only be a couple of choices in the rare earth iridates that give the necessary FM coupling between Ising moments on the rare earth sublattice (Ho and Dy) while also having an insulating ground state and ordered Ir moments. This is a subjective point of view, however, and I don’t feel that any changes to the manuscript necessarily need to be made regarding this.

Answer: The aim of our article was first to introduce a new mechanism that produces the fragmentation of the magnetization in a pyrochlore system and to show that it is physically realized in a pyrochlore compound. The tunable parameter here is the staggered magnetic field that can be simply varied theoretically, which has allowed us to span the H_{loc}/T phase diagram of the fragmented phase. From the experimental point of view, we agree with the referee that this might be more difficult to achieve in a continuous way. However, we believe that it could be attempted through pressure or chemical substitution.

In order to make clearer the main message of our article, we have removed the references to the control and manipulation of monopoles in the abstract of the revised version.

As noted by the referee, we mention at the end of the article the equivalence of our model with the case of the collinear AFM ordered pyrochlore with uniaxial Ising spins in a uniform external field. In this model, the Ising spins all point along the same direction of the lattice. It is thus different from the AIAO case where the spins point along different local directions. (We have replaced “uniaxial” by “collinear” in the text to be clearer on this point). In that collinear case, the phase diagram could be tuned experimentally by varying continuously the external magnetic field. However, to our knowledge, no pyrochlore compounds exhibit such properties (uniaxial anisotropy + antiferromagnetic interactions). It is the reason why we address this question with the multiaxial system and a staggered field.

(2) From a clarity standpoint, I think it would be better to use capital letters for the abbreviations like all-in-all-out (aiao). Otherwise, the word aiao which appears several times could be misunderstood as a typo or some unusual word unknown to the readers. AIAO would make it clear it is an acronym.

Answer: We have corrected it, as well as in other abbreviations concerning “in” and “out” (3I1O, 2I2O, etc...). It has also been corrected in the figures.

(3) In figure 1, on the left side of panel c), it is not clear whether the x-axis relates to temperature (as it does on the right side). The colors fading out seem to suggest this, but it is not labelled on that axis. Some text relating to this point would be helpful.

Answer: It seems that the left panel of Figure 1 c) was not clear enough. The x-axis does not refer to temperature. The goal of this panel was to show that, in the intermediate fragmented phase, where $2 < h_{\text{loc}}/\mathcal{J}_{\text{eff}} < 6$, the spin configuration “3I1O–1I3O” (orange) can be described as the sum of a divergence free (blue) and a divergence full (yellow) part. We have slightly changed the figure to avoid any confusion, and added in the caption “at $T = 0$ ” to make clear that there is no temperature dependence in this panel.

(4) Figure 2 b) shows the ordered magnetic moment as a function of temperature in Ho₂Ti₂O₇, extracted from neutron powder diffraction. The authors argue there is a slight overshoot in the data around 1K, which is consistent with the theoretical model. This is not actually clear in the data, especially since there are no errorbars on those points. It could easily be a plateau. The errorbars should be added, and the wording that describes the possible overshoot should be softened.

Answer: The experimental error bars are within the symbols. We forgot to mention it and it has been added in the caption now. We agree that although one can see the overshoot, it could also be a plateau. For this reason, we have suppressed the words “with a slight overshoot” in the main text, especially since it has no importance on the physical conclusions of this paragraph.

(5) In Figure 3, the specific frequencies of the ac susceptibility data need to be indicated in some way (as a legend, labels directly on the plot, or in the caption with specific start and end points as well as the step size in f).

Answer: We have done it in the caption (labels directly on the plot were not clear enough).

(6) small points: Is the instrument called G41 or G4.1? It is labelled as G41 in the inset of Fig. 2, but in at least two places in the text it is called G4.1. Secondly, in the methods section the word gaz should be gas.

Answer: This has been corrected. G4.1 is the correct spelling.

Reviewers' Comments:

Reviewer #1 (Remarks to the Author):

The authors have satisfactorily addressed referees' comments. I recommend the publication of this paper.

Reviewer #2 (Remarks to the Author):

The authors have addressed my concerns by making some changes to the manuscript, and have corrected my understanding in one case (specifically, concerning the equivalence of this model to the AFM collinear model in a magnetic field, rather than the AIAO model as I had originally thought they meant).

I am happy to recommend publication of this manuscript in Nature Communications.